# Nuclear–Cytoplasmic Shuttling of the Usher Syndrome 1G Protein SANS Differs from Its Paralog ANKS4B

**DOI:** 10.3390/cells13221855

**Published:** 2024-11-08

**Authors:** Jacques S. Fritze, Felizitas F. Stiehler, Uwe Wolfrum

**Affiliations:** Institute of Molecular Physiology, Johannes Gutenberg University Mainz, 55128 Mainz, Germany; jfritze@uni-mainz.de (J.S.F.); f.stiehler@imb-mainz.de (F.F.S.)

**Keywords:** Usher syndrome, USH1G, SANS, ANKS4B, karyopherin, nuclear–cytoplasmic shuttling

## Abstract

The *USH1G* protein SANS is a small multifunctional scaffold protein. It is involved in several different cellular processes, such as intracellular transport, in the cytoplasm, or splicing of pre-mRNA, in the cell nucleus. Here, we aimed to gain insight into the regulation of the subcellular localization and the nuclear–cytoplasmic shuttling of SANS and its paralog ANKS4B, not yet reported in the nucleus. We identified karyopherins mediating the nuclear import and export by screening the nuclear interactome of SANS. Sequence analyses predicted in silico evolutionarily conserved nuclear localization sequences (NLSs) and nuclear export sequences (NESs) in SANS, but only NESs in ANKS4B, which are suitable for karyopherin binding. Quantifying the nuclear–cytoplasmic localization of wild-type SANS and NLS/NES mutants, we experimentally confirmed in silico predicted NLS and NES functioning in the nuclear–cytoplasmic shuttling in situ in cells. The comparison of SANS and its paralog ANKS4B revealed substantial differences in the interaction with the nuclear splicing protein PRPF31 and in their nuclear localization. Finally, our results on pathogenic USH1G/SANS mutants suggest that the loss of NLSs and NESs and thereby the ability to control nuclear–cytoplasmic shuttling is disease-relevant.

## 1. Introduction

SANS (scaffold protein containing ankyrin repeats and SAM domain) is encoded by the *USH1G* gene [1]. Pathogenic variants of *USH1G* lead to the human Usher syndrome (USH), the most common form of combined hereditary deaf-blindness in humans [2,3]. SANS is a small 52 kDa protein that consists of 461 amino acids. It is composed of three N-terminal ankyrin repeats (ANK1-3), a central domain (CENT), a sterile alpha motif (SAM), and a PDZ-binding motif at the C-terminal end (Figure 1A) [1]. Based on its different functional properties the CENT can be further divided into three subdomains, CENTn1, CENTn2, and CENTc [4]. Context-dependent binding of numerous proteins to its different domains characterizes SANS as a potent scaffold protein [5,6,7,8,9,10,11,12]. In auditory hair cells, SANS plays a crucial role in arranging the four other USH type-1 proteins in the mechanosensitive tip-link complex of the stereocilia [13].

In photoreceptor cells of the retina, SANS functions have been associated with ciliogenesis and the intracellular transport of cargoes to their photosensitive outer segment [6,7,8,9,10,11]. In addition to these cytoplasm-associated processes, SANS has also been found in the nucleus [7,12,14]. In the nucleus, SANS has been recently linked to the control of pre-mRNA splicing and the activation of the spliceosome by interacting with several spliceosomal components [4,12]. However, the question of how the distribution of SANS between the two cellular compartments the cytoplasm and the nucleus is regulated in the cell remained open.

Interestingly, SANS has a paralog protein in ANKS4B (ankyrin repeat and SAM domain-containing protein 4B) [14]. Both scaffold proteins are composed of the same principal domain structure (Figure 1A); possess high, up to 80%, amino acid sequence homology; and share common binding partners, such as the USH1C protein harmonin [5,14,15,16]. ANKS4B and SANS are both located at the tips of microvilli, the brush border microvilli of intestinal enterocytes and the stereocilia, highly modified microvilli of inner ear hair cells, respectively [13,17]. However, while both are found in the cytoplasm, ANKS4B has not yet been reported in the nucleus [14,18]. Moreover, the mechanisms underlying this difference in nuclear–cytoplasmic partitioning have yet to be determined and are one of the questions we address here.

Nuclear transport is a fundamental cellular process that regulates the localization of macromolecules within the nuclear or cytoplasmic compartments. In humans, approximately 60 proteins participate in nuclear transport, including Ran system proteins that ensure directed and rapid transport, components that form nuclear pore complexes (NPCs) in the nuclear membrane, and karyopherins that transport cargoes through the NPCs [19]. For bidirectional shuttling of nuclear cargo proteins, karyopherins, namely importins and exportins, are needed [20]. These bind specifically to linear elements in the nuclear cargos, which are called nuclear localization sequences (NLSs) and nuclear export sequences (NESs), respectively.

In the present study, we aimed to shed light on the nuclear–cytoplasmic shuttling of SANS and its paralog ANKS4B. We screened the SANS interactome to identify importins and exportins and used in silico prediction tools to find conserved NLSs and NESs in SANS protein sequences of several vertebrates. We validated the nuclear–cytoplasmic shuttling potential of wild-type and mutated NLS/NES versions of SANS in HeLa and HEK293T cells by quantifying the nuclear–cytoplasmic localization, which confirmed predicted NLSs and NESs to be important for correct nuclear–cytoplasmic shuttling of SANS. In contrast, strict localization of the SANS paralog ANKS4B in the cytoplasm is ensured in the absence of any NLS by several NESs. Finally, we provide evidence that the dysregulation and disruption of SANS’s nuclear–cytoplasmic shuttling are relevant for the development of the USH disease in USH1G patients.

## 2. Materials and Methods

### 2.1. DNA Constructs and Antibodies

DNA Constructs (Table 1), PCR Primers (Table 2) and Antibodies (Table 3) shown below.

### 2.2. Cloning

NLS/NES mutants were created by Q5^®^ Site-Directed Mutagenesis Kit according to the manufacturer’s protocol (NEB, Frankfurt am Main, Germany, #E0554S) with pDONR-SANS as a template. pDONR-ANKS4B was kindly provided by Katja Luck [21]. Gateway^TM^ LR reaction into the appropriate destination vector was performed according to the manufacturer’s protocol (ThermoFisher, Darmstadt, Germany, #11791020).

### 2.3. Cell Lines and Culture

Dulbecco’s modified Eagle’s medium (DMEM) (ThermoFisher, Darmstadt, Germany, #31966021) containing 10% heat-inactivated fetal calf serum (FCS) (Cytiva, Freiburg, Germany, #SV30160.03) was used to culture HeLa (kindly provided by AG Dormann, IMP, Mainz, Germany) and HEK293T cells (ATCC: CRL-3216).

### 2.4. Cell Transfections

HeLa cells and HEK293T cells were seeded on coverslips in 6-well plates with a density of 250,000 cells/well. Cells were transfected with tagged constructs using Lipofectamine^TM^ LTX as described by the manufacturer (ThermoFisher, Darmstadt, Germany, #15338100) and incubated for 24 h. Protein expression levels of the constructs were compared in Western blots (Appendix A). CRM1 cells were treated with 5 nM Leptomycin-B (Selleck Chemicals GmbH, Munich, Germany, #S7580) for the inhibition of the exportin or 40 µM Importazole (Merck, Darmstadt, Germany, #401105) for the inhibition of importins for 4 h before fixation.

### 2.5. siRNA-Mediated Knockdown

For siRNA-mediated knockdowns, we used siRNA for PRPF31 (IDT, Leuven, Belgien, #TriFECTa^®^Kit DsiRNA Duplex, 5′-AGCUAUGGGAUAGUAAGAUGUUUGC-3′) with previously validated knockdown efficiency by qPCR [12]. The siRNA was co-transfected with eYFP-SANS into HeLa cells using Lipofectamine^TM^ RNAiMAX and incubated for 24 h as described by the manufacturer (ThermoFisher, Darmstadt, Germany, #13778075). The siRNA transfection was subsequently detected by qPCR.

### 2.6. Live Cell Imaging and Fluorescence Recovery After Photobleaching (FRAP)

HeLa cells were seeded in an Ibidi 4-well plate (Ibidi, Gräfelfing, Germany #80426) with a density of 40,000 cells/well. Cells were transfected with the fluorescently tagged constructs using Lipofectamine^TM^ LTX. Living cells were analyzed with a Nikon Yokogawa spinning disc microscope with a 60× water objective at 37 °C and imaged every second. FRAP was performed with a 514 nm laser at 36% laser intensity for 100 iterations. Recovery was observed every second for up to 4 min. Quantification was performed with Fiji and EasyFRAP-web [22]. Data are presented as Full-Scale Normalization, and T-half was calculated with Double curve fitting.

### 2.7. Immunocytochemistry

Cells were fixed with 4% paraformaldehyde in PBS for 10 min at RT, washed with PBS, permeabilized with PBST (0.1% Triton-X100 (Roth, Karlsruhe, Germany, # 3051.1)) for 5 min at RT, and blocked with 0.1% ovalbumin and 0.5% fish gelatine in PBS for 1 h at RT. Primary antibodies were incubated overnight at 4 °C, followed by washing with PBS and secondary antibody in a blocking solution containing DAPI (1 mg/mL) (Roth, Karlsruhe, #6335.1) with incubation for 1 h at RT. After washing, samples were mounted in Mowiol (Roth, Karlsruhe, #0713.2).

### 2.8. Cytochemical Staining of Aggresomes

HeLa cells were transfected with eYFP-SANS and NLS/NES mutants with Lipofectamine^TM^ LTX as described by the manufacturer (ThermoFisher, Darmstadt, Germany, #15338100). Control cells were treated 18 h before fixation with DMSO as negative control or with the proteasomal inhibitor MG-132 (5 µM) in DMSO to induce the formation of aggresomes. Cells were fixed with 4% paraformaldehyde in PBS for 10 min at RT, washed with PBS, and permeabilized with PBST (0.1% Triton-X100 (Roth, Karlsruhe, # 3051.1)) 5 min at RT. Subsequently, cells were fluorescently stained for aggresomes with a commercial kit following the supplier’s protocol (Enzo, Lausen, Germany, #ENZ-51035-K100).

### 2.9. Fluorescence Microscopy Analysis

Fixed cells were observed with a Leica TCS SP5 or Zeiss LSM900 (Leica, Wetzlar, Germany). Images were analyzed with Fiji [23] (https://imagej.net/software/fiji/downloads, accessed 1 April 2022). Z-projections were performed over the indicated region of interest with the Fiji Plugin Orthogonal Views. Pearson coefficient calculation was performed with the Fiji Plugin Coloc 2.

### 2.10. Western Blot Analysis

Cells were lysed in Triton-X-100 lysis buffer (50 mM Tris/HCl, 150 mM NaCl, 0.5% Triton-X-100, pH 7.4) containing complete protease inhibitor cocktail (Roche Diagnostics, Mannheim, Germany #04693132001) by sonication. Protein content was quantified using a BCA protein assay (Merck, Darmstadt, #B9643, #C2284) and subjected to SDS-PAGE. The proteins were transferred to a polyvinylidene difluoride (PVDF) membrane (Merck, Darmstadt, #IPVH00010). After blotting, membranes were blocked in AppliChem blocking reagent (AppliChem, Darmstadt, Germany) for 1 h and subsequently incubated with primary antibodies overnight at 4 °C followed by appropriate secondary antibodies. Blots were scanned employing the Odyssey infrared imaging system (LI-COR Biosciences, Lincoln, NE, USA).

### 2.11. In Silico Prediction

The amino acid sequence of the protein of interest was used from uniprot. The following sequences were used: *Homo sapiens* SANS (Q495M9); *Macaca nemestrina* SANS (A0A2K6BVI9); *Mus musculus* SANS (Q80T11); *Danio rerio* SANS (A0A8M1RQW2); *Xenopus laevis* SANS (A0A8J0TLE4); *H. sapiens* ANKS4B (Q8N8V4). NLSs were predicted with a given amino acid sequence by NLStradamus [24]. NESs were predicted with a given amino acid sequence by LocNES [25]. The homology of SANS NLS/NES was aligned with Blastp (https://blast.ncbi.nlm.nih.gov/Blast.cgi?PAGE =Proteins, accessed on 2 April 2024). The homology of SANS and ANKS4B was calculated from the alignment of Blastp (https://blast.ncbi.nlm.nih.gov/Blast.cgi?PAGE =Proteins, accessed on 1 March 2024).

Missense3D or Missense3D-PPI was used as a web-based version (http://missense3d.bc.ic.ac.uk/, accessed on 30 September 2024). For Missense3D, the AlphaFold2-based structure of SANS was used (AF-Q495M9-F1-v4). For Missense3D-PPI, the previously described structures of SANS-PRPF31 and SANS-harmonin were used [4]. The predicted SANS-harmonin is in line with the previously described complex structure [15].

### 2.12. CellProfiler-Based Nuclear/Cytoplasmic Ratio Quantification

Images were preanalyzed with Fiji and then analyzed automatically with CellProfiler [26]. For analysis, we used the pipeline “Human cells” (https://cellprofiler-examples.s3.amazonaws.com/ExampleHuman.zip, accessed on 1 January 2023) with adjustments available on Appendix A (Github: https://github.com/LabWolfrum/Fritze_et_al_2024_SANS_Nuclear_localization).

In short, three tif files (protein of interest, DAPI, Brightfield) were used as input. Primary and secondary structures were identified with CellProfiler, and the mean intensity of the eYFP channel of the nucleus and cytoplasm was used for further statistical analysis.

### 2.13. Cell Fractionation Assay

Cells were seeded in 6-well plates with a density of 250,000 cells/well. Cells were transfected with Lipofectamine^TM^ LTX as described by the manufacturer (ThermoFisher, Darmstadt, Germany, #15338100) and incubated for 24 h. Cells were harvested and lysed into nuclear and cytoplasmic fractions with the commercial kit NE-PER^TM^ (ThermoFisher, Darmstadt, Germany, #78833). Protein amount was measured with a BCA assay, and proteins were eluted with 5x Laemmli buffer and separated by SDS-PAGE, followed by total protein measurement (Revert^TM^ 700 Total Protein Staining) (LI-COR, Lincoln, NE, USA, #926-11010) and Western blotting. Relative intensity was calculated with Fiji for the GFP (cross-reaction for eYFP) band and normalized to total protein and the HistoneH3 band.

### 2.14. Fluorescence Resonance Energy Transfer (FRET) Acceptor Photobleaching Assay

HeLa cells were seeded in 6-well plates with a density of 250,000 cells/well. Cells were transfected with Lipofectamine^TM^ LTX as described by the manufacturer (ThermoFisher, Darmstadt, Germany, #15338100). Fixed HeLa cells were analyzed with a Leica TCS SP8, and FRET acceptor photobleaching was performed following the Leica protocol (https://downloads.leica-microsystems.com/TCS%20SP8/Application%20Note/FRET_AB_with%20SP8-AppLetter_EN.pdf, accessed 1 September 2022). eCFP was used as a donor (D), and eYFP as an acceptor (A). The acceptor was bleached at 100% laser intensity for 10 repeats. FRET efficiency was calculated via $FRET_{eff}=\left(\frac{\left(D_{post}-D_{pre}\right)}{D_{post}}\right)$. To exclude the cellular structure, the mean of six regions of interest (ROIs) in the bleached area was calculated, and the FRET efficiency of an unbleached region (≥3 µm away from bleached ROI) was subtracted from the mean. Bleach efficiency was calculated via the formula $Bleach_{eff}=\left(\frac{1-A\_post}{A\_pre}\right)\times 100$. Only ROIs with > 60% bleach efficiency were used for analysis. Data were normalized to 1 by the positive control eCFP-c-eYFP.

### 2.15. SANS Nuclear Interactome Analysis

SANS nuclear interactome analysis in HEK293T cells [12] was used as input for the Cytoscape plugin ClueGO according to gene names based on HGNC. Gene Ontology (GO) term enrichment analysis was performed by ClueGO v2.3.3.

### 2.16. Statistics

Statistical analysis was performed with R-Studio [27]. The statistical methods are listed in corresponding individual legends. Results are shown from at least 3 separate experiments. Significance was determined as follows: * *p* ≤ 0.05, ** *p* ≤ 0.009, *** *p* ≤ 0.0009.

## 3. Results

### 3.1. Subcellular Localization of SANS

Recent work indicated that SANS shuttles from the cytoplasm into the nucleus [7,12,14]. We confirmed cytoplasmic and nuclear localization of endogenous SANS in HeLa cells by immunocytochemistry using anti-SANS antibodies (Figure 1B) and in HeLa cells transfected with eYFP-SANS and 3xFLAG-SANS monitoring eYFP fluorescence and indirect anti-FLAG immunofluorescence, respectively (Figure 1C). Confocal microscopy also showed that endogenous SANS and both transfected SANS constructs are not homogeneously distributed in the cytoplasm and the nucleus but appear as speckles of different sizes (Figure 1B,C). To verify the nature of these SANS speckles, we performed FRAP (fluorescence recovery after photobleaching) experiments, live cell imaging, and cytochemical staining for aggresomes in eYFP-SANS-expressing HeLa cells (Appendix A). FRAP analysis revealed that eYFP-SANS fluorescence recovered in a fast manner; after less than ~39 sec, almost 50% of its original intensity was recovered, indicating a high mobile fraction (Appendix A). In live cell imaging experiments, we observed fission and fusion of eYFP-SANS speckles on the order of a few seconds (Appendix A). Both the high mobile fraction in FRAP and the fission–fusion properties of eYFP-SANS speckles indicate liquid–liquid phase separation (LLPS) behavior [28].

In addition, we stained eYFP-SANS speckles for aggresomes by applying a cytochemical aggresome staining kit (Appendix A). We did not observe fluorescent aggresome staining of the eYFP-SANS speckles (Appendix A), which contrasted with aggresome-positive controls (Appendix A). Taken together, our analyses indicate that the eYFP-SANS speckles do not represent aggresomes, but rather are formed by LLPS, which is consistent with previous observations for Venus-SANS in HeLa cells [14].

In addition, we noticed a high variance in the subcellular localization of eYFP-SANS in the transfected HeLa cells. While in most cells, eYFP-SANS was mainly found in the nucleus (Figure 1C), there were also cells in which it was rather localized in the cytoplasm (Appendix A). Quantification of eYFP-SANS fluorescence in cytoplasmic and nuclear compartments of over 225 cells by CellProfiler revealed that approximately 30% of SANS was present in the cytoplasm and 70% in the nucleus in HeLa cells (Figure 1D). Quantitative analysis of the compartmental distribution in eYFP-SANS-expressing HEK293T cells confirmed that most SANS (77%) was localized in the nucleus (Appendix A). Next, we biochemically validated the compartmental distribution of SANS by cytoplasmic–nuclear cell fractionations in eYFP-SANS-transfected HeLa cells (Figure 1E,F). Western blot analysis of the nuclear and cytoplasmic fractions showed that approximately 70% of SANS was present in the nucleus, confirming the obtained in situ data.

Taken together our experimental data consistently demonstrated that SANS is predominantly localized in the nucleus.

### 3.2. Identification of Karyopherins in the Nuclear Interactome of SANS

To identify potential karyopherins that bind to SANS, we screened the previously published nuclear interactome of SANS in HEK293T cells [12] for importins and exportins. In the GO-term gene clusters “nuclear import” (GO:0051170) and “nuclear export” (GO:0051168) (Cytoscape plugin ClueGO, https://apps.cytoscape.org/apps/cluego, accessed on 6 June 2024; Appendix A), we identified eight nuclear importins and three exportins including exportin-1/CRM1, the main nuclear protein exporter (Table 4).

Subsequently, we investigated their role by applying commercially available importin and exportin inhibitors to eYFP-SANS-transfected HeLa cells followed by confocal microscopy analysis and quantification with CellProfiler [26]. Applying the importin-β inhibitor Importazole (IPZ) to transfected HeLa had no effect on eYFP-SANS subcellular localization (Appendix A). Further, we investigated the role of the exportin CRM1 in SANS nuclear export by treating eYFP-SANS HeLa cells with the CRM1 inhibitor Leptomycin-B (LMB) [33]. Confocal microscopy and CellProfiler-based quantification of the subcellular localization of SANS revealed that applications of LMB to the cells resulted in a significant increase in the nuclear localization of SANS when compared to the untreated control (Figure 2A,B). In contrast, treatments with the LMB solvent DMSO had no effect on the nuclear localization. Combined, the treatment revealed that the core export of SANS is mediated by CRM1.

### 3.3. In Silico Prediction and Evolutionary Conservation of Nuclear Localization Sequences (NLSs) and Nuclear Export Sequences (NESs) in SANS

For nuclear transport, karyopherins bind to nuclear localization sequences (NLSs) and nuclear export sequences (NESs) present in the cargo molecules [20]. To identify NLSs and NESs in the human SANS sequence, we used the prediction tools NLStradamus [24] and LocNES [25], respectively (Table 5). In addition, the predicted consensus sequences were scored from 0 to 1.

NLStradamus predicted two NLSs within SANS, located in the CENTn2 (NLS_1^213–224^) and SAM domains (NLS_2^436–447^), with scores of 0.733 and 0.679, respectively, both above the NLS threshold of 0.6 [24] (Figure 3A). LocNES identified three NESs within SANS, two in the CENTn2 domain (NES_1^181–195^ and NES_2^235–249^) and one in the SAM domain (NES_3^406–420^), with scores ranging from 0.250 to 0.462, all above the NES threshold of 0.1 [25] (Figure 3A). In addition, we classified the predicted NESs into distinct classes of NES motifs binding to the exportin CRM1 (Table 2).

Sequence alignments of the identified NLS and NES motifs of SANS across five diverse vertebrates from humans to frogs, namely *H. sapiens*, *M. nemestrina*, *M. musculus*, *D. rerio*, and *X. laevis* revealed high to medium conservation of the motifs (Figure 3B–F). Notably, *D. rerio* and *X. laevis* have two and three *Ush1g* genes encoding SANS, respectively. However, all sequences of the predicted NLSs and NESs in the multiple genes of the respective species are identical.

Both NLSs are 100% conserved throughout all the three mammals. In NLS_1^213–224^ of *D. rerio* and *X. laevis*, we observed exchanges of two positively charged amino acids, namely arginine to lysine. However, in NLS_2^436–447^ of both species, exchanges between charged and uncharged amino acids were found.

The CRM1-consensus motif of NES_3^406–420^ is 100% conserved throughout all vertebrate species tested. In contrast, NES_1^181–195^ was conserved in all vertebrates except for *X. laevis*. NES_2^235–249^ was also conserved in all tested species but differed in its starting residue in *M. musculus*, *D. rerio*, and *X. laevis*. Taken together, in silico analyses predicted two NLSs and three NES/CRM1 motifs in SANS that differ in their probabilities and evolutionary conservation in vertebrates.

### 3.4. Analysis of the Nuclear Localization Efficiency of Predicted NLSs in SANS in the Cell

Next, we examined the efficiency of the two predicted NLSs of SANS in cells. Using site-specific mutagenesis, we mutated positively charged residues to uncharged or negatively charged residues in the NLSs of human SANS that should prevent karyopherin binding to the mutated NLS (Table 5, Figure 4A). eYFP-tagged wild-type SANS and SANS NLS mutants (SANS^K213E^/ΔNLS_1^213–224^, SANS^R447W^/ΔNLS_2^436–447^) were transfected into HeLa cells (Figure 4B) and HEK293T cells (Appendix A), and the subcellular distribution of the SANS variants was then assessed by confocal microscopy. Confocal images and their z-projections of eYFP-tagged SANS^K213E^ and SANS^R447W^ indicated an increase in SANS in the cytoplasm of HeLa and HEK293T cells. The speckles formed by the SANS-NLS mutants were not stained with the commercial aggresome staining kit (Appendix A), indicating that they are not aggresomes.

CellProfiler-based quantification of the SANS localization showed a significant decrease in nuclear localization of SANS^K213E^ (ΔNLS_1^213–224^) compared to wild-type SANS in both HeLa and HEK293T cells (Figure 4C and Appendix A). Analysis of SANS^R447W^ (ΔNLS_2^436–447^) showed a decrease in its nuclear localization that was not significant in HeLa cells but was significant in HEK293T cells (Figure 4C and Appendix A). In summary, our findings indicate that NLS_1^213–224^ has a stronger role in the nuclear localization of SANS than NLS_2^436–447^.

### 3.5. Analysis of the Nuclear Export Efficiency of Predicted NESs in SANS in the Cell

We examined the nuclear export efficiency of the predicted NESs of SANS in the cell. We designed and generated SANS NES mutants by exchanging hydrophobic residue leucine (L) for glutamic acid (E) by site-specific mutagenesis for SANS NES_1^181–195^ and SANS NES_2^235–249^ (Table 5, Figure 5A). This resulted in SANS^L195E^ (ΔNES_1^181–195^) and SANS^L249E^ (ΔNES_2^235–249^), which were no longer predicted as NESs by LocNES.

The two NES mutants were tagged with eYFP, transfected into HeLa and HEK293T cells, and examined by confocal microscopy (Figure 5B and Appendix A). Unexpectedly, eYFP-SANS^L249E^ formed prominent cytoplasmic speckles, while eYFP-SANS^L195E^ predominantly localized in small nuclear droplets, as shown by z-projection. The speckles formed by the SANS-NES mutants were not stained with the commercial aggresome staining kit (Appendix A), indicating that they are also not aggresomes. CellProfiler-based quantifications showed no significant difference in the nuclear–cytoplasmic ratio of eYFP-SANS^L249E^ (ΔNES_2^235–249^) compared with eYFP-SANS (Figure 5C and Appendix A). In contrast, the quantification for eYFP-SANS^L195E^ (ΔNES_1^181–195^) exhibited a significant increase in the nuclear localization compared to eYFP-SANS. These findings suggest that NES_1^181–195^ but not NES_2^235–249^ plays a role in SANS nuclear export.

Unfortunately, we failed to design any synthetic mutant for SANS NES_3^406–420^ since all genetic modifications introduced by site-specific mutagenesis in this region led to the emergence of a new NES in near proximity. Therefore, we decided to evaluate the role of SANS NES_3^406–420^ in deletion constructs lacking the domains containing NES_3^406–420^. The confirmed pathogenic frameshift variants of the human *USH1G* gene, SANS^S278Pfs*71^ and SANS^S243*^ (www.LOVD.nl/USH1G, accessed on 4 January 2023), which result in a premature stop at the end of CENTn2 or CENTc domain of SANS, respectively, fulfill this criterion (Figure 6A). Confocal microscopy of eYFP-tagged SANS^S278Pfs*71^ and SANS^S243*^ expressed in HeLa and HEK293T cells revealed enrichments of the pathogenic variants in the nucleus when compared to wild-type SANS, as indicated by z-projection (Figure 6B and Appendix A). CellProfiler-based quantifications confirmed the significant enrichment of both pathogenic variants in the nucleus compared to eYFP-SANS (Figure 6C and Appendix A). The retention of SANS^S278Pfs*71^ and SANS^S243*^ in the nucleus indicates that NES_3^406–420^ plays an additional pivotal role in SANS nuclear export.

In addition, we investigated the cellular localization of the frameshift mutation SANS^V132Gfs*3^, which lacks almost the entire CENTn1 and all following C-terminal domains including all NLSs and NESs (Figure 6A). eYFP-tagged SANS^V132Gfs*3^ localized predominantly in the nucleus of HeLa and HEK293T cells, as indicated by z-projection (Figure 6B and Appendix A). CellProfiler-based quantifications confirmed the significant enrichment of SANS^V132Gfs*3^ in the nucleus compared to eYFP-SANS (Figure 6C and Appendix A).

Taken together, our findings suggest that NES_1^181–195^ and NES_3^406–420^ but not NES_2^235–249^ play a role in SANS nuclear export. Furthermore, the present results on SANS pathogenic variants also suggest that alterations in nuclear shuttling may be part of the pathomechanisms leading to USH1G.

### 3.6. Effcts of NLS and NES Mutations of SANS on the Binding to Compartment-Specific Interaction Partners

To fulfill its diverse functions specific for the different compartments in the cell, SANS interacts with compartment-specific interaction partners; for example, in the cytoplasm with scaffold protein harmonin and in the nucleus with splicing factor PRPF31 [4,6,12,34]. Both interacting partners bind to different domains in SANS. While harmonin binds to the SAM-PBM domain, PRPF31 binds to the CENTn1 domain [4,12,15] (Figure 7A). Both SANS^K213E^ (ΔNLS_1) and SANS^L195E^ (ΔNES_1) are located within the CENTn2 domain and should not directly affect the binding of harmonin or PRPF31. However, these mutations may potentially alter the overall structure of SANS, which may influence these interactions. To investigate this, we used the in silico prediction tool Missense3D [35]. Missense3D predicted no structural changes upon mutation of SANS^K213E^ nor SANS^L195E^ (Appendix A). In addition, Missense3D-PPI, a pipeline for structural changes in protein–protein interfaces [36], did not predict any structural surface alterations to the previously modeled SANS-PRPF31 and SANS-harmonin complexes (Appendix A) [4].

Next, we tested the effects of the SANS^K213E^ (ΔNLS_1) and SANS^L195E^ (ΔNES_1) mutants on their binary interactions with harmonin and PRPF31 in the nucleus of the cell by using fluorescence resonance energy transfer (FRET) acceptor photobleaching assays (Figure 7B) [4]. We co-transfected HeLa cells with eYFP-SANS, eYFP-SANS^K213E^, or eYFP-SANS^L195E^ and harmonin-eCFP or eCFP-PRPF31, respectively, and determined FRET signals in the nucleus, normalized to 1 by the positive control eCFP-c-eYFP. The FRET pairs eYFP-SANS-harmonin-eCFP and eYFP-SANS^K213E^-harmonin-eCFP generated normalized FRET signals of the same height, significantly higher than those of the pair eYFP-SANS^L195E^-harmonin-eCFP (Figure 7C). In contrast, the FRET pairs eYFP-SANS-eCFP-PRPF31 and eYFP-SANS^L195E^-eCFP-PRPF31 generated normalized FRET signals of the same height, significantly higher than those of eYFP-SANS^K213E^-eCFP-PRPF31.

Taken together, the ΔNES and ΔNLS SANS mutants still have the potential to bind to SANS binding partners. However, in the cell, ΔNES SANS sticks in the nucleus and is no longer available together with harmonin for its cytoplasmic functions, and ΔNLS_1 SANS cannot interact with PRPF31 in the nucleus for a spliceosome function because it is not targeted into the nucleus.

### 3.7. SANS and Its Paralog ANKS4B Are Localized to Different Subcellular Compartments

SANS and its paralog ANKS4B have nearly the same principal domain structure (Figure 1A), possess a homology of up to 80% in their amino acid sequences, and share common binding partners, such as USH1C/harmonin [5,14,15,16]. Since the differences in cellular function of closely related proteins often depend on the different spatial arrangements in the cell, we tested whether SANS and ANKS4B differ in their cytoplasmic/nuclear localization.

To determine the subcellular localization of ANKS4B, we analyzed eYFP-tagged ANKS4B in comparison to eYFP-SANS-expressing HeLa cells (Figure 8). Confocal microscopy showed that eYFP-ANKS4B was mainly found in the cytoplasm (Figure 8A), but also occasionally to a lesser extent in the nucleus (Figure 8B). CellProfiler-based quantification demonstrated significantly lower nuclear localization of eYFP-ANKS4B compared to eYFP-SANS (Figure 8C).

NLStradamus predicted no NLS for ANKS4B. However, LocNES predicted six distinct NESs of various CRM1-binding motif classes for ANKS4B spanning its SAM-domain (Appendix A) and predicted scores higher than 0.1, the threshold of LocNES [24].

To test whether these motifs are relevant for the subcellular localization, we treated eYFP-ANKS4B-transfected HeLa cells with the CRM1 inhibitor LMB (Figure 8D). CellProfiler-based quantification demonstrated that the LMB treatment resulted in a significant increase in eYFP-ANKS4B in the nucleus, which was not observed in DMSO-treated control cells (Figure 8E).

Taken together, in contrast to SANS, the SANS paralog ANKS4B is predominantly localized in the cytoplasm guaranteed by nuclear export based on CRM1 exportins.

### 3.8. SANS and Its Paralog ANKS4B Differentially Interact with PRPF31

Next, we tested whether SANS and ANKS4B also differ in their binary interaction with the splicing molecule PRPF31 by FRET acceptor photobleaching assays. For this we co-transfected HeLa cells with eYFP-SANS or eYFP-ANKS4B, respectively, and eCFP-PRPF31. As controls, we used eYFP alone and harmonin-eCFP, which was previously shown to bind to ANKS4B and SANS [5,17]. FRET signals were normalized to 1 by the positive control eCFP-c-eYFP. The FRET pair eYFP-SANS-eCFP-PRPF31 generated significantly higher normalized FRET signals than the control pair eYFP-eCFP-PRPF31, confirming the binary binding of SANS and PRPF31 (Figure 9A) [4,12]. In contrast, the eYFP-ANKS4B-eCFP-PRPF31 FRET pair showed no increase in normalized FRET efficiencies when compared to the control eYFP-eCFP-PRPF31 pair. The height of the normalized FRET signals of eYFP-ANKS4B-harmonin-eCFP was in the range of the FRET signals of the FRET pair eYFP-SANS-eCFP-PRPF31 and significantly higher than the signals of the eYFP-eCFP-PRPF31 control pair, confirming the interaction of ANKS4B with harmonin as previously reported [17].

Next, we co-expressed eYFP-SANS or eYFP-ANKS4B with mRFP-PRPF31 or harmonin-mCherry, respectively, in HeLa cells for subsequent confocal microscopy analyses (Figure 9B–E). Our analyses revealed that ANKS4B co-localized with harmonin in small droplets in the cytoplasm as observed before (Figure 9D) [14], but did not colocalize with PRPF31 in the nucleus, strengthened by the negative Pearson coefficient (R = −0.14) (Figure 9E). In contrast, SANS co-localized with both harmonin and PRPF31, but in different compartments, namely in large speckles in the cytoplasm and in the nucleus, respectively (Figure 9B,C). Both co-localizations were supported by high positive Pearson coefficients (R = 0.96 and 0.92, respectively). This finding suggested that SANS nuclear–cytoplasmic shuttling is additionally controlled by the interaction with partner proteins. However, siRNA-based knockdown of endogenous PRPF31 showed no alteration of eYFP-SANS in HeLa cells (Appendix A), which might be due to the low expression of the endogenous PRPF31.

We also like to note that there are differences in the size of the speckles of eYFP-SANS/harmonin-mCherry and eYFP-ANKS4B/harmonin-mCherry in the cytoplasm (Figure 9B,D). This is probably since the two eYFP-tagged proteins are differentially expressed (Appendix A): the more highly expressed eYFP-SANS possibly recruits more harmonin-mCherry molecules, and correspondingly larger speckles are formed.

In summary, our findings underscore the divergent nuclear localization/export sequences between SANS and the ANKS4B paralog resulting in differential subcellular localization patterns and cellular functions.

## 4. Discussion

The protein traffic across the nuclear pores is mostly mediated by members of the karyopherin-β (or Kap) family commonly known as importins and exportins which specifically recognize NLSs and NESs of the cargo molecules [20]. In the present study, we provide several lines of evidence that the nuclear localization of SANS is regulated by an interplay of importins and exportins. We identified several importins and exportins in the previously described nuclear interactome of SANS [12]. In addition, confocal microscopy showed that Leptomycin-B, a specific inhibitor of CRM1 export, blocks the nuclear export of SANS. Mechanistically, Leptomycin-B binds covalently to the NES binding cleft of CRM1 and thereby competes with the cargo, in this case, SANS, for binding to CRM1 [37]. This also confirms the molecular interaction of SANS with CRM1 as we expected from our SANS interactome data. In silico predictions revealed evolutionary conserved NLSs and NESs in the SANS sequences allowing the binding of importins and exportins. We experimentally confirmed the function of in silico predicted NLSs and NESs by quantifying the subcellular localization of SANS mutants in situ.

Multiple NLSs and NESs have been previously described for nuclear proteins, such as Fanconi anemia group A protein (FANCA) and enzyme 5-lipoxygenase (5-LO), respectively [38,39]. Our data on the two NLSs suggest that NLS_1^213–224^ localized in the central domain of SANS is the major NLS for the nuclear localization of SANS. In comparison to the C-terminal NLS_2^436–447^, NLS_1^213–224^ scored higher in the predictions [24]. In addition, sequence alignments across vertebrates suggest that NLS_2^436–447^ is less conserved. In NLS_2^436–447^, charged amino acids are exchanged for uncharged in lower vertebrates, probably affecting the interaction with importins. Importin binding is based on interactions with positively charged amino acids in NLSs of nuclear proteins [20], and exchanges to uncharged residues have been shown to alter NLS binding to importins [40]. In our study, we also experimentally induced such modifications by site-specific mutations of the NLSs in SANS. The quantitative data obtained showed that the decrease in nuclear localization of SANS with a mutated NLS_1^213–224^ was highly significant in both HeLa and HEK293T cells. In contrast, mutated NLS_2^436–447^ did not lead to a significant decrease in the nuclear localization of SANS in HeLa cells and only to a decrease in lower significance in HEK293T cells. The observed differences between the cell lines in the distribution of proteins between the nucleus and cytoplasm are consistent with previous studies relating them to differences in metabolism between the cell lines [41,42,43].

Based on these data, we hypothesize that importins preferentially bind to NLS_1^213–224^ of SANS for the import into the nucleus, e.g., to accomplish SANS’s nuclear role in pre-mRNA splicing. Interestingly, NLS_1^213–224^ is present in the central domain of SANS, which represents the major binding site for numerous SANS-interacting proteins [7,10,12]. Thus, the binding of cytoplasmic proteins such as myomegalin and whirlin to the central domain [7,10] is likely to compete with the binding of importins to the NLS_1^213–224^. The competition in binding between importins and other cytoplasmic binding partners as a mechanism controlling the nuclear–cytoplasmic shuttling of nuclear proteins has been described previously [44,45]. Accordingly, the binding competition of cytoplasmic interaction partners of SANS and importins to the NLS sites could guarantee its cytoplasmic localization and thus ensure its cytoplasmic functions, e.g., in intracellular transport or ciliogenesis [7,9,11].

In addition, the binding of nuclear proteins may also facilitate nuclear import in a process known as “piggybacking” [46,47]. Since SANS was not completely removed from the nucleus when NLS_1^213–224^ was mutated, and the overexpression of PRPF31 resulted in almost complete nuclear localization of co-expressed eYFP-SANS, we speculated that PRPF31 may play a piggyback role for SANS during nuclear import. However, since siRNA-mediated silencing of endogenous PRPF31 did not alter the nuclear–cytoplasmic shuttling of SANS, we rejected this hypothesis. Alternatively, binding of importins to NLS_2^436–447^ or binding of the identified importins, found in the SANS interactome, to non-classical NLSs, which are not predictable by prediction tools, may compensate for an absence of NLS_1^213–224^.

Inhibition of the nuclear export by LMB suggests CRM1-dependent nuclear export of SANS. The analysis of the three predicted NESs of SANS suggests NES_1^181–195^ of the CENTn2 domain and NES_3^406–420^ of the SAM domain as the major NESs for the nuclear–cytoplasmic shuttling of SANS. All three predicted NESs are above the threshold for qualifying NESs and are highly conserved, except for NES_2^235–249^, which differs in three of five vertebrates in the first residue. As expected for potent NESs, the site-specific mutations resulting in a non-functional NES_1^181–195^ and the deletion variants of SANS lacking the NES_3^406–420^ led to a highly significant increase in their nuclear localization in HeLa and HEK293T cells. In contrast, mutated NES_2^235–249^ did not lead to a significant increase in its nuclear localization. Interestingly, NES_1^181–195^ and NES_3^406–420^ of SANS belong to different classes of CRM1 motifs that can differ in the binding affinity of CRM1 [48] and, thereby, probably also in the nuclear export efficiency.

In comparison to SANS, its paralog ANSK4B was far less abundantly localized in the nucleus, which correlates with the lack of any known nuclear function of ANKS4B [14,17]. At first glance, the absence of ANKS4B might be caused because its sequence does not contain an NLS since none of the predicted tools applied predicted an NLS for importin binding in ANKS4B, but several NESs suitable for CRM1 exportin binding. The equivalent region of ANKS4B to SANS CENTn2, where the potent NLS_1^213–224^ is localized, lacks any NLS but shows a cryptic apical targeting sequence that directs ANKS4B to the microvilli at the apical membrane of epithelial cells [18]. This is further evidence that the two scaffolds have partially diverged in their molecular composition to form unique properties for the cell.

Interestingly, after inhibition of CRM1, more than 80% of the ANKS4B was located in the nucleus. This raises the question of how ANKS4B shuttles into the nucleus. In comparison to SANS (~51.5 kDa https://www.uniprot.org/uniprotkb/Q495M9/entry, accessed 1 October 2023), ANKS4B (~46 kDa https://www.uniprot.org/uniprotkb/Q8N8V4/entry, accesses 15 February 2024) is smaller, in the range of size for passive transfer through the nuclear pore complex commonly reported as between 40 and 50 kDa [19]. However, we also found that the much larger eYFP-tagged ANKS4B ~75 kDa, which we monitored in our experiments, localized in the nucleus. A possible reason for this is that this size exclusion for free diffusion through the NPC is not as effective as assumed. Indeed, more recent findings indicate that larger macromolecules of >100 kDa can also diffuse through the NPC [49]. Alternatively, as known for other proteins and already discussed above for SANS the nuclear import of ANKS4B may also be mediated by non-canonical NLSs, nonlinear sequences, which cannot be predicted by prediction tools [20,50]. Although we could not fully elucidate the regulation of ANKS4B import into the nucleus, our results confirm the previously shown predominantly cytoplasmic localization of ANKS4B. However, once ANKS4B enters the nucleus, it is immediately exported back into the cytoplasm mediated by the CRM1 exportin. This mechanism ensures that ANKS4B is available for its primary functions in the cytoplasm.

In our present study, we found evidence that the shuttling of SANS and ANSK4B between the nucleus and cytoplasm is coordinated by the selective binding of karyopherins to NLSs and NESs and determines their compartment-specific function. In the cell, protein functions and their regulations are commonly fine-tuned by post-translational modifications such as site-specific reversible phosphorylation [43,51,52]. Therefore, it is not surprising that reversible phosphorylation also emerges as an important process for regulating the nuclear availability of proteins [53]. Interestingly, we previously demonstrated that the kinase inhibitor D-ribofuranosylbenzimidazole (DRB) increases the abundance of SANS in the nucleus due to the inhibition of the phosphorylation at S422 (see Appendix A in [9]). S422 is a CK2 (casein kinase 2) phosphosite near NES_3^406–420^. Therefore, it is tempting to speculate that CK2-mediated phosphorylation not only inhibits MAGI2 from binding to the internal PDZ-binding motif (PBM) in SANS-SAM [9] but may also regulate the binding of CRM1 to the NES.

Present confocal microscopy revealed that endogenous SANS and overexpressed tagged SANS form speckles in both the cytoplasm and the nucleus. In our study, we demonstrated that SANS speckles are not insoluble protein aggregates like aggresomes, but rather show features of protein condensates that arise through LLPS: the condensates are not stained for aggresomes; unlike stable aggregates, they are dynamic, and their SANS fraction is highly mobile; furthermore, the overexpression of interacting proteins of SANS, namely of harmonin and PRPF31, increases the size of SANS condensates. These results all suggest that the SANS speckles are formed by LLPS [54,55], which is consistent with previously reported findings [10,12,14]. In these condensates, SANS is probably serving as a scaffold through its multivalent interactions that drive LLPS and organize SANS-containing membrane-less organelles such as Cajal bodies and nuclear speckles or at the base of primary cilia in LLPS [6,7,8,9,10,11,12,34]. LLPS-driven condensates are in a dynamic exchange with their environment, and their components can easily be released and re-integrated [55]. Accordingly, SANS monomers or dimers in the cell could, for example, be easily released from the speckles into the cytoplasm to be transported into the nucleus and reintegrated into nuclear membrane-less organelles and vice versa.

While for the SANS-encoding *USH1G* gene, more than 70 pathogenic variants have been identified thus far as leading to deaf-blindness in patients [56] (www.LOVD.nl/USH1G, accessed on 4 January 2023), to our knowledge, defects in ANKS4B have not been associated with a disease [18]. We have previously demonstrated that pathogenic variants of USH1G/SANS variants can lead to the disruption of fundamental cellular processes in both the cytoplasm and the nucleus: while USH1G/SANS variants lead to altered ciliogenesis and intracellular transport in the cytoplasm [10,11], they cause defects in pre-mRNA splicing in the nucleus [12]. Here, we show that these SANS variants are highly enriched in the nucleus due to the defect in nuclear–cytoplasmic shuttling. It is probable that the lack of NESs leads to defects in nuclear export.

Although the SANS variants are targeted into the nucleus, they cannot fulfill their functions in the splicing process there, as we have previously demonstrated [12]. It is possible that not all interaction partners that interact with SANS during proper activation of the spliceosome can interact with the mutated, truncated versions of the scaffold protein. Probably more importantly, due to its pathogenic mutations, SANS is no longer available for its functions in the cytoplasm. A combination of defects in possibly unbalanced cytoplasmic trafficking and ciliogenesis as well as in pre-RNA splicing in the nucleus may underlie the pathology leading to USH1G. This complex scenario may be present in the retinal cells in the eye. In the inner ear, USH1G-associated deafness is more likely caused by defects in the assembly of mechanosensitive tip-link complex in the stereocilia of auditory hair cells [13].

## 5. Conclusions

USH1G/SANS is a multifunctional scaffold protein that participates in various cellular processes executed in different compartments of the cell, namely in the cytoplasm or nucleus. To fulfill its specific tasks, efficient and controlled nuclear–cytoplasmic shuttling of the SANS protein is mediated by karyopherins binding to NLSs and NESs in the SANS. This is very different from its strict cytoplasmic paralog ANKS4B, in which several NESs guarantee its localization and function in the cytoplasm. Finally, we provide evidence that the dysregulation and disruption of SANS’s nuclear–cytoplasmic shuttling can be relevant for the development of the USH disease in USH1G patients.

## Figures and Tables

**Figure 1 cells-13-01855-f001:**
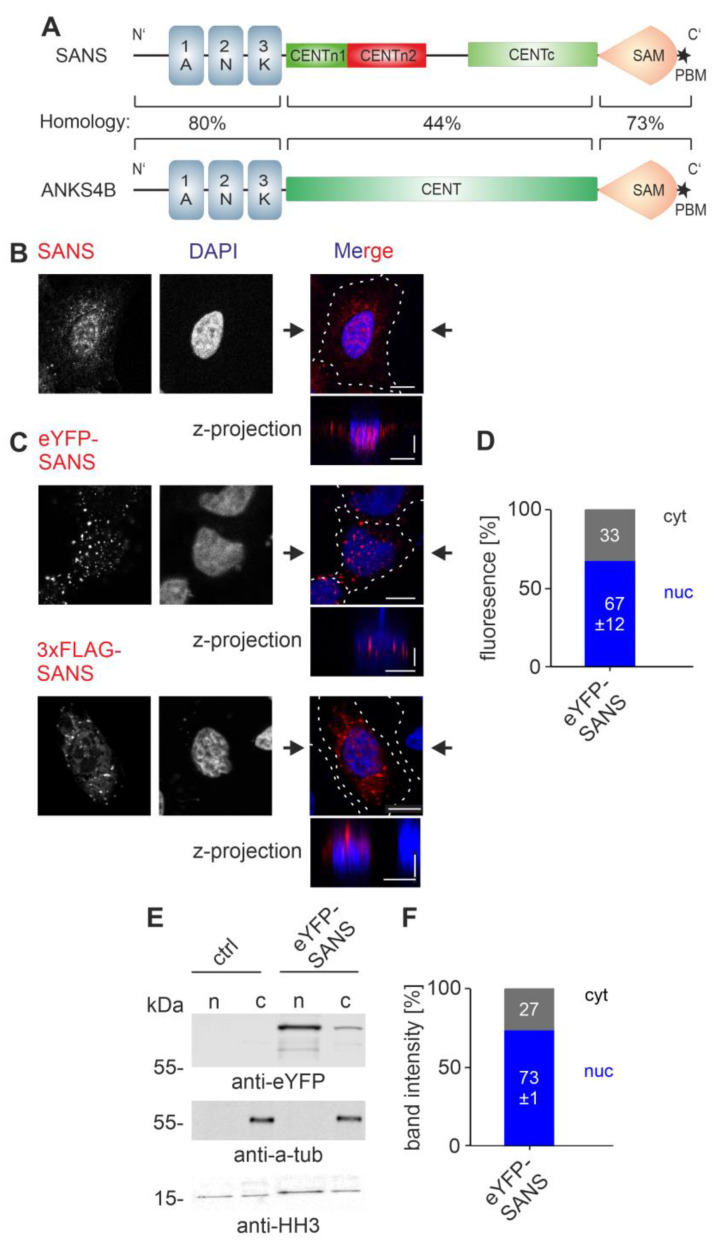
Nuclear localization of SANS. (**A**) Domain structure of SANS and its paralog ANKS4B: Both consist of three ankyrin repeats (ANK1-3), a central domain (CENT), divided in SANS into three parts (CENTn1 (green), CENTn2 (red), and CENTc (green)), a sterile alpha motif (SAM), and a C-terminal type-I PDZ-binding motif (PBM =asterisk). ANKS4B and SANS amino acid sequences are highly similar in their N- and C-terminal regions. (**B**,**C**) Confocal microscopy of HeLa cells either stained for endogenous SANS (**B**) or transfected with eYFP-SANS, 3xFLAG-SANS (**C**), counterstained with DAPI. Endogenous SANS and 3xFLAG-SANS were visualized by indirect immunofluorescence of anti-SANS and anti-FLAG. (**B**,**C**) Endogenous SANS, eYFP-SANS, and 3xFLAG-SANS were localized in the cytoplasm and the nucleus. (**D**) Quantification of eYFP-SANS localization by CellProfiler v4.2.6 in HeLa cells. eYFP-SANS is localized in both the nucleus and cytoplasm. (**E**) Western blot analysis of eYFP SANS in the nuclear (n) and cytoplasmic (c) fraction of eYFP-SANS-transfected HeLa cells using anti-eYFP, anti-tubulin (cytoplasmic housekeeping protein), and anti-HH3 (nuclear housekeeping protein). In each lane, 30 µg of the total protein was loaded. (**F**) Quantification of the eYFP-SANS bands in the nuclear (n) and cytoplasmic fraction (c). Band intensity was normalized to the total protein amount. The qualifications of the microscopic and biochemical analysis consistently demonstrate that the majority of eYFP-SANS (~70%) is localized in the nucleus. Black arrows: position of Z-projections. Scale bars: horizontal = 10 µm; vertical = 2 µm. Data show mean values ± standard deviation from three independent experiments.

**Figure 2 cells-13-01855-f002:**
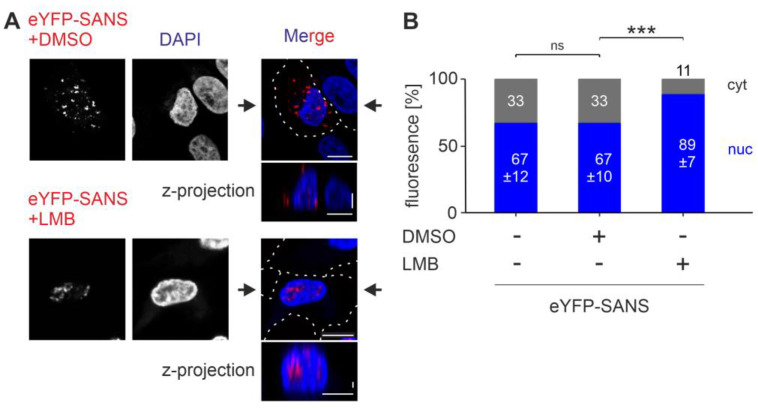
Nuclear localization of SANS in HeLa after treatment with the exportin CRM1 inhibitor Leptomycin-B. (**A**) Confocal microscopy of HeLa cells transfected with eYFP-SANS (red) and treated with 5 nM CRM1 inhibitor Leptomycin-B (LMB) or its solvent DMSO. (**B**) Quantification of (A) by CellProfiler. eYFP-SANS was significantly enriched in the nucleus after LMB treatment. Black arrows: position of Z-projections. Scale bars: horizontal = 10 µm; vertical = 2 µm. Data show mean values ± standard deviation from three independent experiments. Student’s *t*-test was performed for three independent experiments with a minimum of 75 cells; ns = not significant; *** = *p* ≤ 0.0009.

**Figure 3 cells-13-01855-f003:**
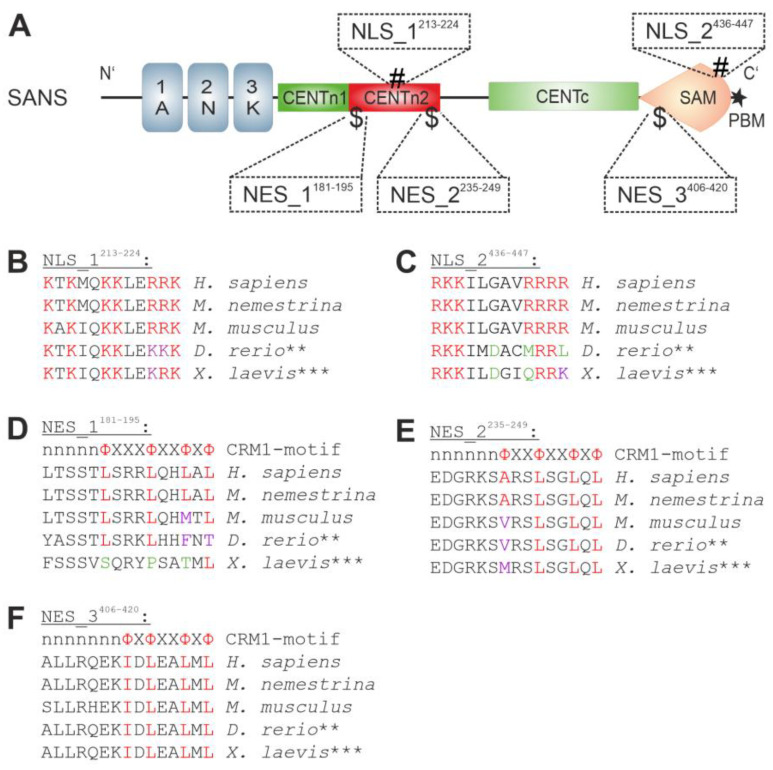
Prediction of nuclear localization sequences (NLSs) and nuclear export sequences (NESs) of SANS. (**A**) Three NESs ($; NES_1^181–195^, NES_2^235–249^, NES_3^406–420^) and two NLSs (#; NLS_1^213–224^, NLS_2^436–447^) were predicted for SANS by NLStradamus and LocNES. (**B**–**F**) Conservation of SANS NLS_1^213–224^ (**B**), NLS_2^436–447^ (**C**), NES_1^181–195^ (**D**), NES_2^235–249^ (**E**), and NES_3^406–420^ (**F**) with blastp in comparison to human SANS for five model organisms. In (**B**,**C**), positively charged residues (red), arginine-to-lysine exchanges (purple), and charge changes (green) are indicated in NLSs. In (**D**–**F**), in the first line, the position of consensus amino acid residues in the CRM1 motif is indicated in red. Valid (purple) and invalid (green) residue changes in the consensus sequence of the CRM1 motif are indicated. ** and *** indicate that the genomes of *D. rerio* and *X. laevis* possess two and three *Ush1g* genes encoding for SANS orthologs, respectively, which do not differ in the NLS and NES sequences.

**Figure 4 cells-13-01855-f004:**
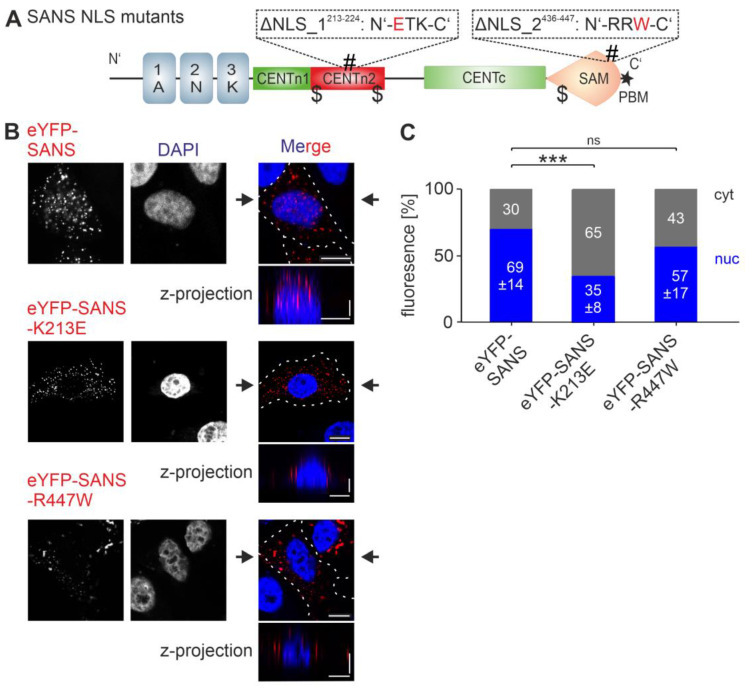
Subcellular localization of SANS NLS mutants. (**A**) Domain structure of SANS; the two NLS (#) mutants K213E (ΔNLS_1^213–224^) and R447W (ΔNLS_2^436–447^) are indicated above the cartoon. (**B**) Confocal microscopy of HeLa cells transfected with eYFP-SANS (red) or SANS NLS mutants, counterstained with DAPI. (**C**) Quantification of (**B**) by CellProfiler. eYFP-SANS^K213E^ differed significantly from eYFP-SANS. Black arrows: position of Z-projections. Scale bars: horizontal = 10 µm; vertical = 2 µm. Data show mean values ± standard deviation from three independent experiments. Student’s *t*-test was performed for three independent experiments with a minimum of 75 cells. ns = not significant; *** = *p* ≤ 0.0009.

**Figure 5 cells-13-01855-f005:**
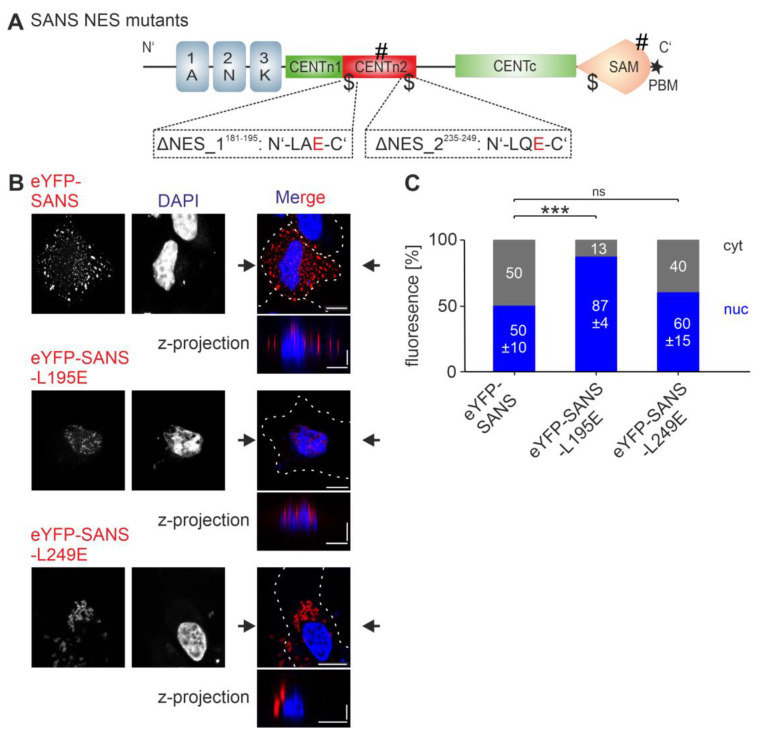
Localization of SANS NES mutants. (**A**) Domain structure of SANS; the two NES ($) mutants L195E (ΔNES_1^181–195^) and L249E (ΔNES_2^235–249^) are indicated below the cartoon. (**B**) Confocal microscopy of HeLa cells transfected with eYFP-SANS (red) or NES mutants, counterstained with DAPI. (**C**) Quantification of (**B**) by CellProfiler. eYFP-SANS^L195E^ was significantly enriched in the nucleus compared to eYFP-SANS. Black arrows: position of Z-projections. Scale bars: horizontal = 10 µm; vertical = 2 µm. Data show mean values ± standard deviation from three independent experiments. Student’s *t*-test was performed for three independent experiments with a minimum of 75 cells. ns = not significant; *** = *p* ≤ 0.0009.

**Figure 6 cells-13-01855-f006:**
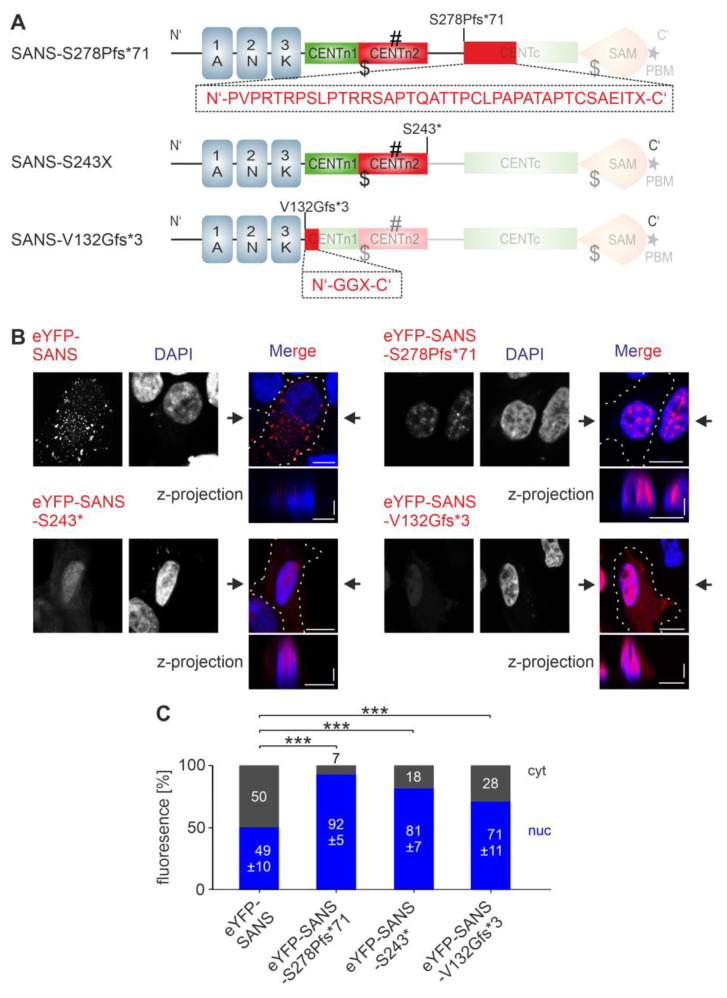
Localization of pathogenic variant of SANS in HeLa cells. (**A**) The pathogenic variant SANS^S278Pfs*71^ and SANS^V132Gfs*3^ are frameshift mutations which lead to missense extension (red boxes and amino acid sequences below) and a premature stop in CENTc or CENTn1, respectively. The pathogenic variant SANS^S243*^ has a premature stop codon after the CENTn2 domain. (**B**) Confocal microscopy of HeLa cells transfected with eYFP-SANS (red), eYFP-SANS^S278Pfs*71^, eYFP-SANS^S243*^ and eYFP-SANS^V132Gfs*3^, counterstained with DAPI. (**C**) Quantification of (**B**) with CellProfiler. All pathogenic variants were significantly enriched in the nucleus. Black arrows: position of z-projections. Scale bars: horizontal = 10 µm; vertical = 2 µm. Data show mean values ± standard deviation from three independent experiments. Student’s *t*-test was performed for three independent experiments with a minimum of 75 cells. *** = *p* ≤ 0.0009.

**Figure 7 cells-13-01855-f007:**
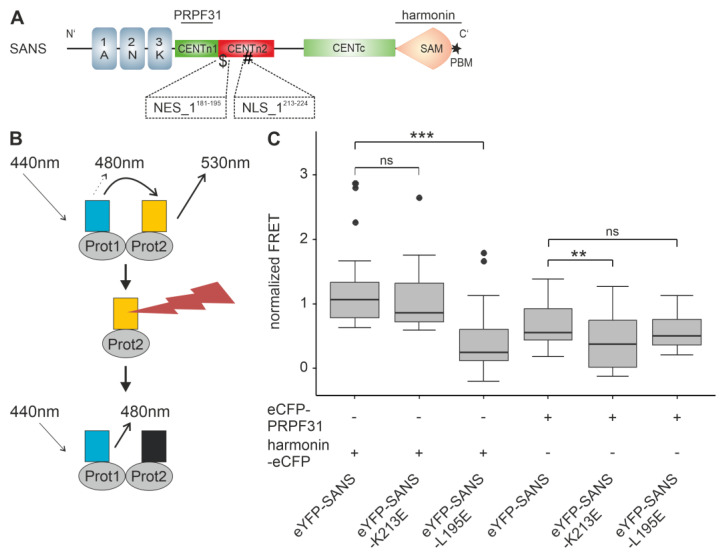
Validation of the interaction of SANS NLS/NES mutants with SANS binding partners fluorescence resonance energy transfer (FRET) in the nuclear compartment. (**A**) Schematic of the SANS domain structure indicating NES1 and NLS1 and the regions for harmonin and PRPF31 binding, black lines in SANS SAM-PBM and CENTn1, respectively. (**B**) Illustration of the FRET acceptor bleach assay. Interaction of two proteins, either eCFP (blue)- or eYFP (yellow)-tagged, leads to FRET (upper). If the acceptor (eYFP) is bleached (flash) (middle), increased emission of the donor (eCFP) can be measured (lower). (**C**) FRET acceptor bleach assay in the nucleus of co-transfected HeLa cells. FRET efficiencies were normalized to 1 by the fused eCFP-c-eYFP FRET pair (positive control). FRET pairs eYFP-SANS^L195E^-harmonin-eCFP and eYFP-SANS^K213E^-eCFP-PRPF31 showed a significant decrease in interaction compared to eYFP-SANS. Outliers are shown as dots above the boxplots. Wilcoxon signed-rank test was performed for three independent experiments. ns = not significant; *** = *p* ≤ 0.0009; ** = *p* ≤ 0.009.

**Figure 8 cells-13-01855-f008:**
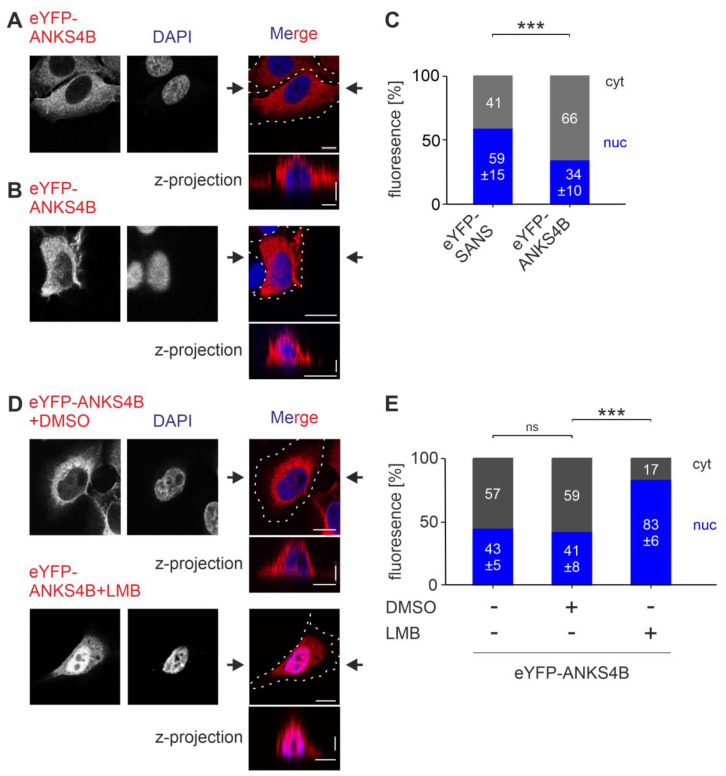
Comparison of the nuclear–cytoplasmic localization of SANS and its paralog ANKS4B in HeLa cells. (**A**,**B**) Confocal microscopy of HeLa cells transfected with eYFP-ANKS4B (red), counterstained with DAPI. (**C**) Quantification with CellProfiler for eYFP-SANS- and eYFP-ANKS4B-transfected cells. eYFP-ANKS4B was highly enriched in the cytoplasm compared to eYFP-SANS. (**D**) Confocal microscopy of HeLa cells transfected with eYFP-ANKS4B (red), counterstained with DAPI. Cells were treated with DMSO or 5 nM Leptomycin-B (LMB), an established inhibitor for CRM1. (**E**) Quantification of (**D**) with CellProfiler. eYFP-ANKS4B was highly enriched in the nucleus after treatment with LMB. Black arrows: position of Z-projections. Scale bars: horizontal = 10 µm; vertical = 2 µm. Data show mean values ± standard deviation from three independent experiments. Student’s *t*-test was performed for 3 independent experiments with a minimum of 75 cells. ns = not significant; *** = *p* ≤ 0.0009.

**Figure 9 cells-13-01855-f009:**
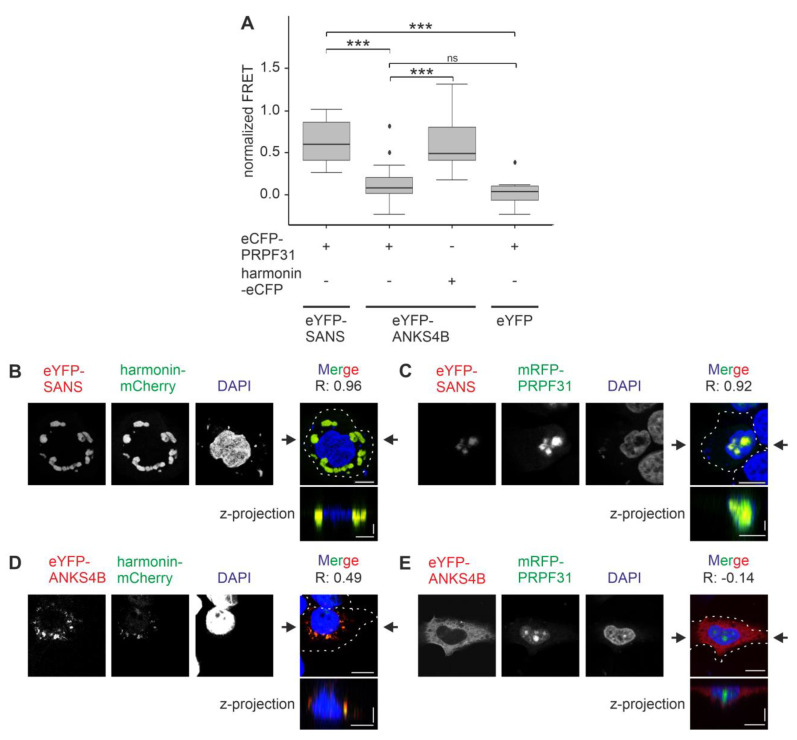
ANKS4B binary interaction in the nucleus. (**A**) FRET assay in co-transfected HeLa cells. FRET efficiencies were normalized to 1 by the fused eCFP-c-eYFP FRET pair (positive control). FRET pair eYFP-ANKS4B-eCFP-PRPF31 does not show a significant increase in the normalized FRET efficiencies when compared to eYFP FRET pair negative controls. Outliers are shown as dots above/below the boxplots. Dunn’s test after a Kruskal–Wallis test was performed for three independent experiments. (**B**–**E**) Confocal microscopy of HeLa cells co-transfected with eYFP-SANS (red, (**B**,**C**)) or eYFP-ANKS4B (red, (**D**,**E**)) and harmonin-mCherry (green) or mRFP-PRPF31 (green), counterstained with DAPI. eYFP-ANKS4B only co-localized with harmonin-mCherry but not with mRFP-PRPF31. Black arrows: position of Z-projections. Scale bars: horizontal = 10 µm; vertical = 2 µm. Pearson coefficient R values indicate co-localization. ns = not significant; *** = *p* ≤ 0.0009.

**Table 1 cells-13-01855-t001:** DNA constructs. Asterisk indicates that the mutations.

DNA Constructs	Reference
eYFP-SANS-S243*	this work
eYFP-SANS-K213E	this work
eYFP-SANS-R447W	this work
eYFP-SANS-L195E	this work
eYFP-SANS-L249E	this work
eYFP-ANKS4B	this work
harmonin-mCherry	this work
eYFP-SANS	[4]
eYFP-SANS-S278Pfs*71	[4]
eYFP-SANS-V132Gfs*3	[4]
eCFP-PRPF31	[4]
harmonin-eCFP	[4]
eCFP-c-eYFP (FRET positive control)	[4]
mRFP-PRPF31	[4]
FLAG-SANS	[12]
pDONR-SANS	[10]
pDONR-SANS-S243*	[10]
pDest-743 (eYFP control)	kindly provided by Ronald Roepman
pDONR-ANKS4B	kindly provided by Katja Luck [21]

**Table 2 cells-13-01855-t002:** PCR Primers.

Primer Name	Sequence 5′-3′
K213E_fwd	GGCGAGACCAAGATGCAG
K231E_rev	CCTGGCCGTGCCGTG
R447W_fwd	CGGTGGCAGGCGATG
R447W_rev	CCTCCTCACGGCCC
L195E_fwd	GCGGAGGGCAGCC
L195E_rev	CAGATGCTGCAGCCGG
L249E_fwd	CAGGAGGGCAGCGAC
L249E_rev	CAGGCCCGAGAGCG
qPCR PRPF31 F	CAGACACAGGTAAACGAGGC
qPCR PRPF31 R	CTGGAGTGGGGTGAAGGC

**Table 3 cells-13-01855-t003:** Antibodies.

Antibody Name	Supplier	Catalog No.
Rabbit-anti-SANS	Proteintech, Planegg-Martinsried, Germany	21936-1-AP
Rabbit-anti-histonH3	Cell Signaling Technology, Danvers, MA, USA	4499
Mouse-anti-α-tubulin	Merck, Darmstadt, Germany	T9026
Mouse-anti-FLAG	Merck, Darmstadt, Germany	F1804
Mouse-anti-GFP (cross-reaction to eYFP)	Proteintech, Planegg-Martinsried, Germany	66002-1-Ig
Rat-anti-RFP	Proteintech, Planegg-Martinsried, Germany	5f8
Rabbit-anti-γ-tubulin	Merck, Darmstadt, Germany	T6557
Donkey-anti-rabbit Alexa488	ThermoFisher, Darmstadt, Germany	A-21206
Donkey-anti-rabbit Alexa680	ThermoFisher, Darmstadt, Germany	A-10043
Donkey-anti-mouse Alexa800	ThermoFisher, Darmstadt, Germany	SA5-10172
Donkey-anti-rat Alexa680	ThermoFisher, Darmstadt, Germany	A-78947

**Table 4 cells-13-01855-t004:** Nuclear exportins and importins identified as potential SANS-interacting proteins in the nucleus [12].

Gene	Protein	Export/Import	NLS/NES Recognition
KPNB1	Importin-β	Protein importer	classic NLS [20] non-classic NLS [20]
IPO4	Importin-4	Protein importer	classic NLS [29] non-classic NLS [20]
IPO5	Importin-5	Protein importer	classic NLS [30] non-classic NLS [20]
IPO7	Importin-7	Protein importer	classic NLS [31] non-classic NLS [20]
IPO8	Importin-8	Protein importer	classic NLS [31] non-classic NLS [20]
IPO9	Importin-9	Protein importer	non-classic NLS [20]
IPO11	Importin-11	Protein importer	non-classic NLS [20]
IPO13	Importin-13	Protein importer	classic NLS [32] non-classic NLS [32]
XPO1	CRM1/Exportin-1	Protein exporter	NES
XPO5	Exportin-5	dsRNA exporter	-
XPOT	Exportin-T	amino-acylated tRNAs export	-

**Table 5 cells-13-01855-t005:** Nuclear localization (NLSs) and export sequences (NESs) of SANS predicted by LocNES and NLStradamus.

NLS/NES	Sequence	Score	CRM1-Class	NLS/NES Mutations
NLS_1^213–224^	213-KTKMQKKLERRK-224	0.733	-	K213E: ETKMQKKLERRK
NLS_2^436–447^	436-RKKILGAVRRRR-447	0.679	-	R447W: RKKILGAVRRRW
NES_1^181–195^	181-LTSSTLSRRLQHLAL-195	0.262	1a	L195E: LTSSTLSRRLQHLAE
NES_2^235–249^	235-EDGRKSARSLSGLQL-249	0.250	1b	L249E: EDGRKSARSLSGLQE
NES_3^406–420^	406-ALLRQEKIDLEALML-420	0.462	2	n.a.

## Data Availability

All raw data are available through contacting the corresponding author.

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
