# Peer review of "Nuclear–Cytoplasmic Shuttling of the Usher Syndrome 1G Protein SANS Differs from Its Paralog ANKS4B"

_cells, 2024, doi:10.3390/cells13221855_

Round 1

Reviewer 1 Report (New Reviewer)

Comments and Suggestions for Authors

This is a study to investigate the molecular basis controlling the nuclear-cytoplasmic shuttling of the protein Sans, a scaffold known to play an essential role in hearing and vision.  Importantly, genetic mutations in Sans result in Usher syndrome Type 1G, a form of deaf-blindness in humans. This manuscript is a logical extension of previous work from the Wolfrum lab that discovered that Sans has an expected role in pre-mRNA processing in the nucleus.  The authors use a bioinformatic approach to identify key NLS and NES sequences in Sans, which are then validated in cell cultured cells using point mutants.  The authors also extend this analysis to look at particular Usher syndrome Type 1G variants that would impact these NLS and NES sequences.  FRET analysis is then used to examine how interactions with either cytoplasmic or nuclear binding partners influences the nuclear/cytoplasmic localization of Sans.  Finally, the authors perform a comparative study of Sans with its paralogue ANKS4B, focusing on whether ANKS4B also is found in the nucleus.

The experiments are well-done, the conclusions are supported by high quality data, and the figures are easy to understand and appropriate for the content.  This work will be of broad interest to people in the field of cell biology, in particular nuclear-cytoplasmic shuttling, mRNA processing, and also Usher syndrome research.

I only have very minor edits/suggestions:

1)    There are a few typos in the manuscript:

·      Line 296: located in the CENTn2 (NLS_1213-

o   Should read: CENTn2 (NLS_1213-

·      Lines 539-540: “In addition, Confocal microscopy showed that eYFP-ANKS4B -B, a specific inhibitor of CRM1 export, blocks the nuclear”

o   Should read “leptomycin-B”

·      Lines 620-621: “is smaller, in the range of the size to Present confocal microscopy revealed passively through the nuclear pore complex….”

o   Needs to be edited.

2)    The authors present convincing data that the NLS in CENTn2 (213-224) plays an important role in nuclear localization of Sans, and that this NLS does not appear to be found in ANKS4B.  The authors might want to note that the equivalent region in ANKS4B is part of an apical targeting sequence for ANKS4B, used to target ANKS4B to microvilli (Graves et al JBC, 2020).  The equivalent region of the Sans 213-224 NLS sequence has been reported as a basic-hydrophobic-basic motif in ANKS4B that helps form a putative membrane binding domain.  This is further evidence that the two scaffolds have partially diverged away from each other, and possess some unique properties compared to each other.

Author Response

Reviewer 2 Report (Previous Reviewer 2)

Comments and Suggestions for Authors

The authors have adequately addressed all my concerns in their revision, producing a greatly improved, robust manuscript.

I noted just one minor typographical error to fix: the two sentences that span lines 620-622 have an insertion in blue font that disrupts the meaning of both sentences. 

Author Response

This manuscript is a resubmission of an earlier submission. The following is a list of the peer review reports and author responses from that submission.

Round 1

Reviewer 1 Report

Comments and Suggestions for Authors

In this study the authors address the distribution of SANS and ANKS4B in the nuclear and cytoplasmic compartments of the cell. They identify putative NLS and NES sequences in addition to the importins and exportins

responsible for the nuclear-cytoplasmic shuttling, which are then experimentally verified. Overall, the manuscript is well written. however, it would benefit from addressing some concerns as below.

The main experimental data that this study relies on, are the fluorescence images and their analysis to indicate the localization of SANS and its mutants. However, there are some concerns- In fig 1C, the OE eYFP SANS clearly looks like it has a lot of cytoplasmic signal- the z project or quantification in fig 1D doesn’t accurately reflect the image shown here. What are these bright puncta in the cytosol? The nuclear/cytoplasmic blot in fig 1E is more convincing- but again, does not reflect what is seen in the representative image in fig 1C. Are these cytoplasmic puncta lost in the initial cell pellet during the fractionation step? For fig 1D, was the entire volume in z project used for calculation? These major discrepancies need to be addressed.

Also, there is a possibility of a cell-to-cell variation and that most of the images did not have the bright cytoplasmic puncta, in which case a better representative image should be added. For example, why does SANS localization in fig S2A look quite different from that in fig 1C- how variable are the levels of OE SANS? When dealing with localization in different cellular compartments of an over-expressed protein, it is crucial to address- the variability in expression and if the overall levels of protein are not the same, it can be mistaken for what looks like a difference in compartmentalization. Moreover, when looking at mutants of a protein of interest, it is important to verify that the expression levels are comparable to the wild type. The NLS mutants in fig 4 seem to have overall lesser expression levels that the wt SANS protein. The authors should show support their claims with a immunoblot to confirm.

The data shown with ANKS4B is convincing, however, there might be some cell-to-cell variability here as well, for examples, why are bottom two images of ANKS4B in fig 8C different- especially when the claim is that the expression of PRPF31 does not affect the localization of ANK? Does the expression of harmonin vary as well and since that itself in different in the top and bottom images of fig 8C, it would explain the discrepancy and should  atleast be addressed in the text.

While looking at the pathogenic variants, the authors claim that the differential localization could be crucial. The manuscript would highly benefit if the authors have a functional assay for SANS that they can show to be altered when the localization is experimentally changed by the NLS/NES mutations as well as for the pathogenic mutants.

Reviewer 2 Report

Comments and Suggestions for Authors

This manuscript investigates the mechanisms underlying nucleocytoplasmic shuttling of SANS, a small multifunctional protein with both cytoplasmic and nuclear roles in human cells. Pathogenic variants that contribute to Usher syndrome, the most common form of combined hereditary deaf-blindness, are included in the study, emphasizing the importance of understanding the trafficking of SANS. The authors used in silico prediction tools to find conserved NLSs and NESs in SANS and then attempted to validate these localization signals in human cells in culture using fluorescence microscopy and biochemical fractionation to quantify nuclear-cytoplasmic localization. They provide some evidence for sequences being "necessary" from import or export, but do not show that these sequences are "sufficient." There are major concerns that need to be addressed to provide evidence in support of the authors conclusions.

1. SANS does not have a homogeneous distribution and, from the representative images provided by the authors, appears to localize in cytoplasmic or nuclear foci/aggregates/speckles of varying sizes (e.g., see Fig. 1). The authors need to discuss what these foci/aggregates might represent. The only time they are mentioned specifically is in the context of one of the NLS mutants (line 318-319), but these aggregates are present the wild-type SANS as well.

2. It is not clear how drawing a line through the cell can serve as an accurate RO1, given that the line may entirely miss bright foci, or only include bright foci, depending on where it is drawn. These same issues with quantification apply to Fig. 2, Fig. 4, and Fig. 5. Only the pathogenic mutant is clearly more nuclear (Fig. 6) and it is interesting to note that this mutant has a homogeneous distribution so can be quantified accurately. In Figs. 4 and 5, the huge aggregates in some of the images look like localization to the aggresome (a cellular compartment at the nuclear periphery where misfolded proteins collect). In Fig. 4, given that one of the SANS NLS mutant is in this "aggresome," it is not surprising that it remains cytoplasmic. Cytoplasmic localization may not be due to an NLS mutation, but rather mislocalization to the aggresome due to protein misfolding. The authors should determine what these aggregates/foci represent, using appropriate markers. 

3. The western blot reporting on quantification of biochemical fractionation in Fig. 1E is also not convincing. Were equal protein amounts loaded, or were equal cell equivalents loaded (to account for the differing volumes of the cytoplasm vs nucleus)?

4. The caption for Fig. 1 mentions a students t-test, but no p values are provided.

5. In Fig. 2, it is surprising that SANS seems to be imported as large aggregates. The authors need to discuss how to resolve their quantification of nuclear localization with the presence of aggregates.

6. Much of the information in Table 2 is repetitious with Fig. 3.

7. Line 299: What is meant by results being "..slightly significant"? Results are either statistically significant or not statistically significant.

8. Line 383: The authors note that no NLS is predicted for ANKS4B. How does it get into the nucleus where it is then trapped by LMB treatment (preventing export)? Testing some domain constructs for NLS activity would be of interest.

9. Table 3 would be more appropriate as supplemental data.

10. The quantification in Fig. 7 is questionable. For ANKS4B the nucleus from the cell image looks like a black hole. Why then does it show up as being 34% nuclear in the bar graph (B)? Showing more representative cell images could help to clarify this apparent discrepancy. The bar graph in part D also does not seem to reflect the images shown.

11. The paper would be strengthened by testing for direct interaction (e.g. by coimmunoprecipitation assays) with SANS of the importins and exportins identified as potential interacting partners by in silico methods. Without this experimental work, line 539 is an overstatement of the results, since selective binding is not shown in this study.